# Learning to Break the Loop: Analyzing and Mitigating Repetitions for Neural Text Generation

**Jin Xu**[1][*] **Xiaojiang Liu**[4]**, Jianhao Yan**[2]**, Deng Cai**[3]**, Huayang Li**[4]**, Jian Li**[1]
[1]Institute for Interdisciplinary Information Sciences, Tsinghua University
[2]School of Engineering, Westlake University
[3]The Chinese University of Hong Kong
[4]Apple
xujin21@mails.tsinghua.edu.cn, xiaojiang_liu@apple.com, elliottyan37@gmail.com
thisisjcykcd@gmail.com, hli46@apple.com, lijian83@mail.tsinghua.edu.cn

## Abstract

While large-scale neural language models, such as GPT2 and BART, have achieved impressive results on various text generation tasks, they tend to get stuck in undesirable sentence-level loops with maximization-based decoding algorithms (*e.g.*, greedy search). This phenomenon is counter-intuitive since there are few consecutive sentence-level repetitions in human corpora (e.g., 0.02% in Wikitext-103). To investigate the underlying reasons for generating consecutive sentence-level repetitions, we study the relationship between the probabilities of the repetitive tokens and their previous repetitions in the context. Through our quantitative experiments, we find that 1) Language models have a preference to repeat the previous sentence; 2) The sentence-level repetitions have a *self-reinforcement effect*: the more times a sentence is repeated in the context, the higher the probability of continuing to generate that sentence; 3) The sentences with higher initial probabilities usually have a stronger self-reinforcement effect. Motivated by our findings, we propose a simple and effective training method **DITTO** (PseuDo-RepetITion PenalizaTiOn), where the model learns to penalize probabilities of sentence-level repetitions from pseudo repetitive data. Although our method is motivated by mitigating repetitions, experiments show that DITTO not only mitigates the repetition issue without sacrificing perplexity, but also achieves better generation quality. Extensive experiments on open-ended text generation (Wikitext-103) and text summarization (CNN/DailyMail) demonstrate the generality and effectiveness of our method. Code is released at `https://github.com/Jxu-Thu/DITTO`.

## 1 Introduction

Recently, large-scale Transformer-based [34] neural language models (e.g., GPT2 [29], BART [17] and OPT [39]) have shown remarkable capability to generate text and achieved better performance than before, such as open-ended generation [29, 5] and summarization tasks [17, 39]. However, models with standard maximization-based decoding are known to get stuck in redundant consecutive repetitions [13]. As shown in Figure 1, the model has a stronger preference for consecutive sentence-level repetitions than the word- or phrase-level while human language has fewer consecutive sentence-level repetitions, which shows a discrepancy between human language and the generated texts. Existing approaches to mitigate repetitions can be categorized into decoding-based and training-based methods. Decoding-based methods rectify these issues by soft or hard $n$-gram blocking [26, 16],

---

[*]The work was conducted in Apple.

**Prefix:** The rim is slightly raised about the surrounding surface and it has an outer rampart that has been only lightly impacted . No significant craters intersect the rim , and it is sloped about 1 @.@ 5 ° toward the direction 50 – 90 ° from the Earth .

**Greedy Decoding:**
The age of the crater is about 3 @.@ 6 billion years and it has been in the proximity of the south lunar pole for at least 10 @,@ 000 years . The South Crater is located on the southern edge of the northern highlands . The South Crater is located on the southern edge of the northern highlands . The South Crater is located on the southern edge of the northern highlands . The South Crater is located on the southern edge of the northern highlands . The South Crater is located on the southern edge of the northern highlands . The South Crater is located on the southern edge of the northern highlands . The South Crater is located on the southern edge of the northern highlands . The South Crater is located on the southern edge of the northern highlands . …

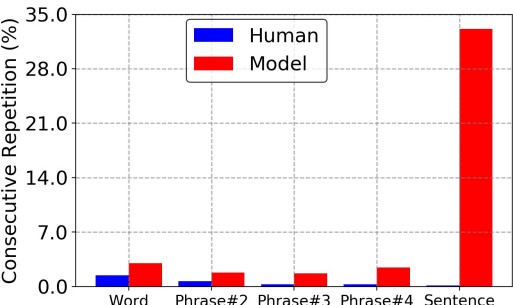

Figure 1: Statistics of human sentences versus model generation on the dev set of Wikitext-103. We train a Transformer model (750M parameters, similar to GPT-2 Large) on large-scale human corpus Wikitext-103 (over 100 million words). **Left**: Greedy decoding gets stuck in consecutive sentence-level repetition. **Right**: The percent of consecutive repetition of the word-, phrase (# number of words)- and sentence-level (see Appendix A for formulations of consecutive repetition). The model results are the average of generated texts given different prefixes from the Wikitext-103 dev set. Specifically, given 50 tokens as the prefix, the model greedily generates the next 200 tokens. Compared to human language, the model has substantially more consecutive *sentence-level* repetition.

perplexity/information controlling [3, 21], and stochastic sampling [7, 14, 29, 13, 35]. Training-based methods minimize the probabilities of tokens that already are generated in the previous context [36, 19]. Despite their effectiveness, the reasons why the model prefers repetitions and how repetition occurs during decoding are still unclear.

Fu *et al*. [9] is the first to analyze the repetition problems from a theoretical perspective by assuming that the language models can be approximated by short-sighted first-order Markov models. However, Holtzman *et al*. [13] observe the cases of a positive feedback loop of repetitions, which indicates that language models do look at the long-distance context and may not be simply viewed as first-order Markov models. The cases also reveal that the probabilities of repetitive tokens have certain relationships with previous repetitions in context. However, they do not analyze why the model prefers consecutive repetitions. In this paper, we further dig into the problem and conduct *quantitative* experiments to analyze the underlying issue of the repetition.

For a quantitative investigation on consecutive repetitions, we compare the probabilities of the same tokens in repetitive sentences. For example, given a sequence, 'I love oranges . I love oranges .', we compare the probability $\mathcal{P}_\theta$('oranges'|'I love oranges . I love') with $\mathcal{P}_\theta$('oranges'|'I love'). The difference between them is that, for the second 'oranges', there is already a token 'oranges' that shares the same *sentence-level context* 'I love'. We manually repeat the sentence $n$ times as the context so that the next 'oranges' has $n$ repetitions in context. In this way, we can investigate the relationship between the probability of the token and the number of repetitions in context. For example, we can first construct the context by repeating the sentence 'I love oranges .' $n$ times plus 'I love', and then obtain the probability that the model outputs 'orange' at the current step.

Through our quantitative investigation across the different corpus, we find that 1) **The model tends to raise the probability of repeating the previous sentence.** Specifically, even if there is only one sentence-level context repetition, the probability of repetition at the current step increases in most cases. The cause of the phenomenon may be that the model is so confident with its context, when there is a previous token (i.e., 'oranges') sharing the same sentence-level context (i.e., 'I love'), that the model learns a shortcut to directly copy the token; 2) **Self-reinforcement effect: the probability of repetition increases almost monotonically with the number of historical repetitions.** Finally, the probability of repetition stabilizes around a certain ceiling value. As shown in Figure 2, as the number of repetitions increases, the probability of the word 'rounds' and 'general' both increase almost monotonically and finally stabilize; 3) **Sentences with higher initial probabilities usually have a stronger self-reinforcement effect.** For example, in Figure 2, we can find that the sentence with a higher initial probability (i.e., the red bar at '0' of x-axis) grows faster and can reach an extremely high value with a few repetitions. Furthermore, the sentences with a higher initial likelihood (e.g., sentences generated by the model itself with maximization-based decoding algorithms) may have a stronger self-reinforcement effect.

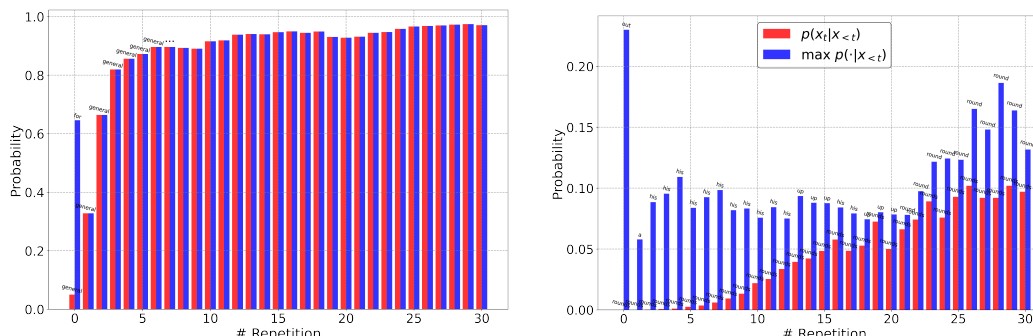

Figure 2: Manually repeat a given sentence several times, feed to the model and observe the same token's probability $p(x_t|x_{<t})$ (in red) and maximum probability $\max p(\cdot|x_{<t})$ (in blue) as *sentence repetition times increase*. **Left**: Repeat a normal sentence ('*She is named a proxy general under Gaara .*') and present the probability of the token 'general'. **Right**: Repeat a random sentence ('*fría backed rounds Manganiello Benzedrine Magruder Crego Stansel Zemin compressus .*') where tokens are randomly sampled from the vocabulary, and present the probability of the token 'rounds'. The probability of repetition (in red) has a self-reinforcement effect: the probability of repetition (*y-axis*) increases almost monotonically with the number of historical repetitions (*x-axis*). Best viewed in color and zoomed in a desktop monitor.

According to our findings, the reasons why the model tends to repeat themselves are as follows: The sentence repetition occurs since the previous sentence generated by the maximization-based decoding algorithms has a relatively high probability, and the model tends to further increase the probability of repeating that sentence. Once the model generates one repetitive sentence, the probability of the repetitive sentence would be further enhanced since there are more repetitions sharing the same sentence-level context to support the choice of copying. As a result, the model gets stuck in the sentence-level loops due to the *self-reinforcement effect*.

To mitigate the tendency to repeat previous sentences and overcome the self-reinforcement effect, we aim to make the model aware that the probabilities of generating redundant sentences should be reduced. In addition, the more sentence-level repetitions exist, the lower the probability of the redundant sentence should be. In this paper, we propose a simple and effective training-based method, PseuDo-RepetITion PenalizaTiOn (**DITTO**). We first manually construct pseudo data by repeating a sentence sampled from the training corpus. Then, we design a repetition penalization loss function on these pseudo data for the model to learn to exponentially decay the repetition probability as the number of sentence repetitions increases. The experimental results show that DITTO not only significantly reduces repetitions without sacrificing language model capability in terms of perplexity, but also achieves better generation quality. In the Wikitext-103 [22] open-ended generation task, our methods can achieve lower perplexity and higher MAUVE scores than the strong competitors. With stochastic decoding strategies such as top-$k$ [7] and nucleus sampling [13], generated texts by our methods are the closest to human-written texts, as measured by MAUVE score and human evaluation. In the commonly-used CNN/DailyMail summarization benchmark [12, 23], our methods also achieve the best performance among the strong competitors.

## 2   Analyzing Repetition

In this section, we would like to quantitatively investigate the relationship between the probabilities of repetitive sentences and previous repetitions in context. We first define several metrics to measure how the probability changes as the number of repetitions increases. Then, we summarize our findings from the experimental results, and discuss how sentence-level repetitions occur during decoding.

### 2.1   Experiment Design

Formally, we have a sentence $\mathbf{s}$ from a corpus $\mathcal{D}$ and repeat it $N$ times to construct a sample sequence $\mathbf{x} = (\mathbf{s}^0, \mathbf{s}^1, \mathbf{s}^2, \cdots, \mathbf{s}^N)$ where $\mathbf{s}^n = (x_{n,1}, \cdots, x_{n,L_\mathbf{s}})$. $x_{n,l}$ is the $l$-th token in the $n$-th repetition of the sentence $\mathbf{s}$ and $L_\mathbf{s}$ is the number of tokens in sentence $\mathbf{s}$. Denote the *sentence-level context* of token $x_{n,l}$ as $\mathbf{x}_{n,<l} = (x_{n,1}, \cdots, x_{n,l-1})$. We define that there is a *sentence-level context repetition* for the

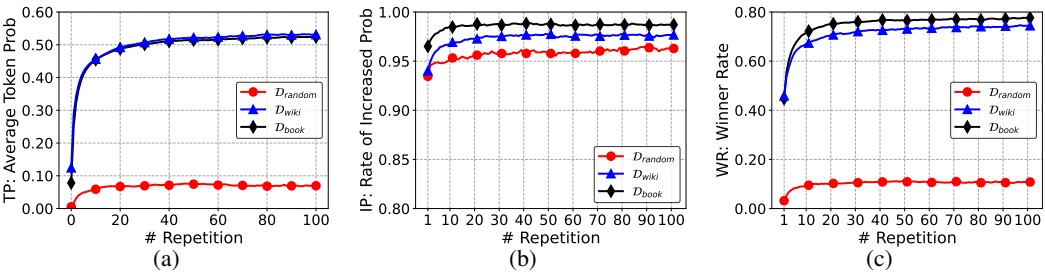

Figure 3: Construct a sample by repeating a sentence $N = 100$ times, manually feed it to the model and calculate $\text{TP}_n$ (average token probability), $\text{IP}_n$ (rate of increased probability) and $\text{WR}_n$ (winner rate) for $n = 1, \cdots, N$. The results average on sentences from different corpus $\mathcal{D}_{\text{random}}$, $\mathcal{D}_{\text{book}}$ and $\mathcal{D}_{\text{wiki}}$ respectively.

token $x_{n,l}$ if and only if there is another same token $x_{i,l}$ in context such that $\mathbf{x}_{i,<l} = \mathbf{x}_{n,<l}$ and $i < n$. For example, for the sequence 'I love oranges . I love oranges .' aforementioned, the second token 'oranges' has a *sentence-level context repetition*. By feeding the sequence $\mathbf{x}$ to a pre-trained model $\mathcal{P}_\theta$, we can obtain the probability distribution $\mathcal{P}_\theta(x_{n,l}|\mathbf{x}_{<n,l})$ where $\mathbf{x}_{<n,l} = (\mathbf{s}^0, \cdots, \mathbf{s}^{n-1}, \mathbf{x}_{n,<l})$. To study whether $\mathcal{P}_\theta(x_{n,l}|\mathbf{x}_{<n,l})$ increases as $n$ increases, we define several metrics as follows:

- **Average Token Probability:** $\text{TP}(\mathbf{s}^\mathbf{n}) = \frac{1}{L_\mathbf{s}} \sum_{l=1}^{L_\mathbf{s}} \mathcal{P}_\theta(x_{n,l}|\mathbf{x}_{<n,l})$, which is the average token probability of $n$-th repetitive sentence $\mathbf{s}^n$. $\text{TP}(\mathbf{s}^\mathbf{0})$ is the initial probability of tokens in the first sentence.

- **Rate of Increased Token Probability**: $\text{IP}(\mathbf{s}^\mathbf{n}) = \frac{1}{L_\mathbf{s}} \sum_{l=1}^{L_\mathbf{s}} \mathbb{1}(\mathcal{P}_\theta(x_{n,l}|\mathbf{x}_{<n,l}) > \mathcal{P}_\theta(x_{0,l}|\mathbf{x}_{<0,l}))$ where $\mathbb{1}$ is the indicator function. We use $\text{IP}(\mathbf{s}^\mathbf{n})$ to calculate that, how many probabilities of tokens increase in $\mathbf{s}^\mathbf{n}$ compared to the initial ones in $\mathbf{s}^\mathbf{0}$.

- **Winner Rate:** We say $x_{n,l}$ is a *winner* if $\mathcal{P}_\theta(x_{n,l}|\mathbf{x}_{<n,l}) > \mathcal{P}_\theta(x_{0,l}|\mathbf{x}_{<0,l})$ and $x_{n,l} = \arg\max \mathcal{P}(\cdot|x_{<n,l})$. Then, we define the winner rate as $\text{WR}(\mathbf{s}^\mathbf{n}) = \frac{1}{L_\mathbf{s}} \sum_{l=1}^{L_\mathbf{s}} \mathbb{1}(x_{n,l} \text{ is a } winner)$. A higher winner rate means that the repetitions are more likely to be generated by a maximization-based decoding algorithm such as greedy decoding.

Following the previous work [36, 19], the pre-trained model is a 16-layer Transformer decoder trained on Wikitext-103 [22]. The details are introduced in Sec. 4.1. Given the corpus $\mathcal{D}$, we can calculate the average values of TP, IP and WR of the $n$-th repetitive sentence $\mathbf{s}^n$ as

$$\text{TP}_n = \frac{1}{|\mathcal{D}|} \sum_{\mathbf{s} \in \mathcal{D}} \text{TP}(\mathbf{s}^n), \quad \text{IP}_n = \frac{1}{|\mathcal{D}|} \sum_{\mathbf{s} \in \mathcal{D}} \text{IP}(\mathbf{s}^n), \quad \text{WR}_n = \frac{1}{|\mathcal{D}|} \sum_{\mathbf{s} \in \mathcal{D}} \text{WR}(\mathbf{s}^n)$$

by enumerating all sentences $\mathbf{s}$ in $\mathcal{D}$. To further study the effect of different corpus, we construct three corpus 1) $\mathcal{D}_{\text{random}}$: the tokens of sentences are randomly sampled from the vocabulary of the model; 2) $\mathcal{D}_{\text{book}}$: the sentences are randomly sampled from BookCorpus [40], and 3) $\mathcal{D}_{\text{wiki}}$: the sentences are randomly sampled from the dev set of Wikitext-103. The size of all different corpus is 1,000.

## 2.2 Results and Analyses

In the analyses, we attempt to answer: 1) Why sentence repetitions occur? 2) Why the model gets stuck into the repetition loop? 3) What kinds of sentences are more likely to be repeated?

**Why sentence repetitions occur?** As shown in Figure 3 (b), $\text{IP}_1$ is higher than 90% across the various corpus, meaning that even if there is only one sentence-level context repetition, the probability of repetition increases in most cases. That indicates *the model has a strong preference to repeat the previous sentence*. Note that the token repetition has not occurred at the current prediction step, and there is only the same sentence-level context. For example, model assigns a higher probability to $\mathcal{P}_\theta(\text{'oranges'}|\text{'I love oranges . I love'})$ than $\mathcal{P}_\theta(\text{'oranges'}|\text{'I love'})$ since the model has seen the pattern 'I love oranges' in the previous context. Thus, the model may be too confident with its previous context repetition and learn a 'cheap' shortcut where it directly copies the next token, 'oranges'. It is quite different from human language because human would try to avoid full sentence repetition, as the repeated sentence has no new information in most cases.

**Why the model gets stuck into the repetition loop?** Furthermore, as shown in Figure 3, as the number of repetitions increases, TP, IP and WR increase monotonically. That means, *the sentence-level repetition has a self-reinforcement effect:* the more times a sentence has been repeated in the context, the higher the probability of continuing to generate that sentence. Finally, IP, WR, TP converge around certain ceiling values.

**What kinds of sentences are more likely to be repeated?** We can find that *the sentences with high initial sentence probability $TP_0$, such as sentences sampled from $\mathcal{D}_{wiki}$ and $\mathcal{D}_{book}$, have a stronger self-reinforcement effect:* TP and WR grows faster and reach the high ceiling values with a few repetitions (See Appendix D.1 for more experiments at sentences with different initial probabilities). The higher TP and WR, the more likely the sentence to be repetitively generated by maximization-based decoding algorithms. Note that sentences generated by a maximization-based decoding algorithm have higher initial likelihoods. Thus, the model prefers to repeat itself.

**Analyses** During decoding with maximization-based algorithms, the sentence-level repetition occurs because 1) Previous generated sentences have high likelihoods and thus have more potential to be repeated; 2) Given previously generated sentences, the model is more likely to generate the repetitive sentence since the model is confident on the historically generated context and tries to find hints in context for the current generation; 3) Once the model repeats the sentence for several times, it would get stuck in the sentence loop due to *self-reinforcement effect*. The effectiveness of stochastic sampling approaches such as top-$k$ [7] and nucleus sampling [13] may rely on 1) compared to the maximization-based decoding algorithms, the likelihood of previously generated sentences is lower and thus the sentences have less risk of being repeated; 2) current tokens are generated stochastically rather than chosen with maximum probability which can avoid the self-reinforcement effect.

## 3 Pseudo-repetition Penalization Training

According to our analyses in Sec. 2, it can be clearly seen that the core issue of sentence-level repetition for the model is the tendency to repeat previous sentences and the self-reinforcement effect. In this section, we propose a simple and effective method, named PseuDo-RepetITion PenalizaTiOn (**DITTO**). It first manually feeds repetitive sentences to the model and then explicitly teaches the model to be averse to such repetitions.

**Pseudo Repetitive Data Construction.** To construct a pseudo repetitive sample $\mathbf{x}$, we first randomly pick a sentence $\mathbf{s}$ from the training corpus. Then, the pseudo data $\mathbf{x} = (\mathbf{s}^0, \cdots, \mathbf{s}^N) = (x_{0,0}, \cdots, x_{1,0}, \cdots, x_{N,0}, \cdots, x_{N,L_s})$ is constructed by repeating the sentence $\mathbf{s}$ by $N+1$ times. The sentences are repeated until they reaches the maximum input sequence length of the model (e.g., 1,536 tokens in the Transformer for open-end generation).

**Sentence-level Repetition Penalization.** To define the per-step penalization loss for token $x \in \{x_{1,0}, \cdots, x_{N,L_s}\}$, we define the training objective for the $l$-th token in the $n$-th repetition of the sentence $\mathbf{s}$ as

$$\mathcal{L}_{\text{DITTO}}^{n,l}(\mathcal{P}_\theta(x_{n,l}|\mathbf{x}_{<n,l})) = -\log(1 - \left|\mathcal{P}_\theta(x_{n,l}|\mathbf{x}_{<n,l}) - \lambda \cdot \mathcal{P}_\theta^*(x_{n-1,l}|\mathbf{x}_{<n-1,l})\right|), \quad (1)$$

where $\mathcal{P}_\theta^*(\cdot)$ means that the value is excluded for gradient backpropgation, which can implemented by *tensor.detach* in pytorch [25]. $\lambda$ is the penalization factor. When $\lambda = 1$, the loss function is minimized when the probability of token $x_{n,l}$ in $n$-th repetition is same as that in the $(n-1)$-th repetition to avoid the self-reinforcement effect; when $\lambda < 1$, the probability of token $x_{n,l}$ in $n$-th repetition should be smaller than that in the $(n-1)$-th repetition to make model averse to sentence-level repetition. In other words, the probability of tokens in repetitive sentence should decay *exponentially* with a factor of $\lambda$, where $\lambda$ is a hyper-parameter.

In our experiments, we apply the sentence-level repetition penalization by fine-tuning a standard MLE baseline. Fine-tuning is done by equally mixing the sentence-level repetition penalization update and normal MLE loss update. We find that, appending the previous context of the repetitive sentence as the prefix to the pseudo repetitive data $\mathbf{x}$ can achieve better and more stable performance. We have tried other alternative loss functions, but they do not perform better. See Appendix C for the details.

**Discussion**   In human language, there are necessary token-level and phrase-level repetitions that naturally occurs. For example, names of people, city and lemma, set phrases and proverbs appear many times in long documents such as Wikipedia. Thus, given the prefix, the model should have the ability to copy these words or phrases from the prefix and increase their repetition probabilities. These useful repetitions can be viewed as *positive samples*, and our pseudo repetitive data can be viewed as *negative samples*. Combining them for training, the model should learn to distinguish between useful and useless contexts. Thus, although our method is motivated by overcoming sentence-level repetition issues, it may improve the learning ability and generalizability of the model. Unlike previous methods (e.g., token-level unlikehood training [36] and $n$-gram blocking [26, 16]) that put hard constraints to penalize any repetitions (regardless of the necessity of repetitions), our training objective explicitly encodes the intuition that the model should be inclined to avoid over-repetitions. Therefore, it may alleviate the drawbacks of over-penalization.

## 4   Experiments

### 4.1   Setup

**Dataset and Model Training**   We train models on the benchmark dataset Wikitext-103 [22] to evaluate the performance of open-ended generation. The dataset contains over 100 million words. The experiments are conducted at the word level. The model architecture and training hyper-parameters exactly follow the implementations of Welleck *et al.* [36][2]. Specifically, we use a 16-layer Transformer with 8 attention heads, hidden size 1024 and fully-connected dimension 4096. There are a total of 750 million parameters, similar to GPT-2 Large [29]. We first train the baseline model with standard maximum likelihood (MLE) for a maximum of 150k updates with a batch size of 3 samples. The maximum length of each training sample is 1,536 tokens. Then, we fine-tune the model that has the best validation perplexity with DITTO training for 10k steps. Unless otherwise mentioned, $\lambda$ is set as 0.5. Baseline models are trained following their official implementations. All models are trained on 8 NVIDIA Tesla V100 GPUs. The experiments are implemented based on fairseq codebase [24].

**Evaluation**   Following common practices [36, 19], we evaluate the quality of open-ended sequence completions on the test set of Wikitext-103, where the prefix is 50 tokens, and models autoregressively generate the next 100 tokens. Models including various baselines are selected with the best perplexity on the validation set for a fair comparison. The evaluation metrics are listed as follows:

- **MAUVE**: MAUVE [27] is a metric to measure how close model-generated text is to human language by approximately calculating the KL divergence between the distribution of model generated sequences and human language. The range of MAUVE is from 0 to 1. A higher MAUVE score indicates better generated sequences.

- **Perplexity and Accuracy**: Given the prefix and true next token, we use perplexity, and next-token prediction accuracy to measure the language modeling ability.

- **Repetition**: Following previous work [36, 19], we calculate the portion of duplicate 4-grams (**Repetition-4**) in a generated sequence to measure phrase-level repetition, defined as $1.0 - |\text{unique 4-grams}|/|\text{4-grams}|$, and average over completions. Similarly, we use portion of duplicate sentences (**Repetition-Sen**) to measure sentence-level repetition, defined as $1.0 - |\text{unique sentences}|/|\text{sentences}|$. Since there are natural repetitions in human language, the optimal model should produce text whose repetition metrics are **close to** that of the gold text.

### 4.2   Results of Open-ended Generation

**Method Comparison**   We compare with training-based methods since decoding-based methods can be readily applied to models with our training method. Following [36, 19], we first compare different training-based algorithms based on greedy decoding. The results are presented in Table 1. Our DITTO can significantly reduce the phrase-level and sentence-level repetitions and achieve the best MAUVE score of 0.77. Although UL-token+seq can generate fewer repetitions, its improvement in the general quality of the model measured by MAUVE is limited since its generations are usually not relevant to the prefix (See Appendix F for examples). Compared to the standard baseline MLE,

---

[2]`https://github.com/facebookresearch/unlikelihood_training`

Table 1: Results of different training-based methods on the test set of Wikitext-103 for open-ended generation. The results are reported based on three runs with different random seeds. The best value is **bolded** and the second best is underlined.

| Model | MAUVE | Perplexity | Accuracy | Repetition-4 | Repetition-Sen |
|-------|-------|------------|----------|--------------|----------------|
| MLE [29] | $0.34_{\pm0.02}$ | $25.68_{\pm0.04}$ | $0.39_{\pm0.00}$ | $44.20_{\pm1.43}\%$ | $14.50_{\pm1.59}\%$ |
| UL-token [36] | $0.57_{\pm0.01}$ | $26.98_{\pm0.12}$ | $0.39_{\pm0.00}$ | $28.30_{\pm0.78}\%$ | $7.40_{\pm0.83}\%$ |
| UL-token+seq [36] | $0.48_{\pm0.03}$ | $25.95_{\pm0.08}$ | $0.40_{\pm0.00}$ | $\mathbf{7.60_{\pm0.46}\%}$ | $\mathbf{0.05_{\pm0.03}\%}$ |
| SG [19] | $\underline{0.74_{\pm0.01}}$ | $25.84_{\pm0.06}$ | $\underline{0.40_{\pm0.00}}$ | $23.00_{\pm0.28}\%$ | $5.24_{\pm0.75}\%$ |
| DITTO (ours) | $\mathbf{0.77_{\pm0.01}}$ | $\mathbf{24.33_{\pm0.04}}$ | $\mathbf{0.42_{\pm0.00}}$ | $\underline{22.00_{\pm0.31}\%}$ | $\underline{2.85_{\pm0.74}\%}$ |
| Human | - | - | - | 1.10% | 0.01% |

Table 2: Results of different training-based methods on the test set of Wikitext-103 under different stochastic decoding algorithms. $k = 50$ and top-$p$ ($p = 0.9$) for nucleus sampling. The numbers *closest* to *human scores* are in **bold** except for MAUVE [27].

| Search | Model | MAUVE | Repetition-4 | Repetition-Sen |
|--------|-------|-------|--------------|----------------|
| Top-$k$ | MLE [29] | $0.94_{\pm0.00}$ | $1.60_{\pm0.09}\%$ | $0.25_{\pm0.06}‰$ |
| | UL-token [36] | $0.95_{\pm0.00}$ | $0.70_{\pm0.13}\%$ | $0.00_{\pm0.00}‰$ |
| | UL-token+seq [36] | $0.93_{\pm0.01}$ | $0.09_{\pm0.11}\%$ | $0.06_{\pm0.02}‰$ |
| | SG [19] | $0.93_{\pm0.01}$ | $0.50_{\pm0.19}\%$ | $0.00_{\pm0.00}‰$ |
| | DITTO | $\mathbf{0.96_{\pm0.00}}$ | $\mathbf{1.00_{\pm0.10}\%}$ | $\mathbf{0.09_{\pm0.01}‰}$ |
| Nucleus | MLE [29] | $0.94_{\pm0.00}$ | $1.40_{\pm0.08}\%$ | $\mathbf{0.08_{\pm0.01}‰}$ |
| | UL-token [36] | $0.94_{\pm0.00}$ | $0.47_{\pm0.08}\%$ | $0.00_{\pm0.00}‰$ |
| | UL-token+seq [36] | $0.94_{\pm0.01}$ | $0.08_{\pm0.05}\%$ | $0.02_{\pm0.02}‰$ |
| | SG [19] | $0.93_{\pm0.01}$ | $0.40_{\pm0.19}\%$ | $0.06_{\pm0.01}‰$ |
| | DITTO | $\mathbf{0.96_{\pm0.00}}$ | $\mathbf{0.98_{\pm0.09}\%}$ | $\mathbf{0.08_{\pm0.01}‰}$ |
| | Human | - | 1.10% | 0.10‰ |

other competitors reduce repetitions by sacrificing perplexity while DITTO achieves lower perplexity and higher accuracy. These results show that DITTO can not only mitigate repetitions but also help the model improve the language modeling and generation quality.

**Compatibility with Stochastic Decoding Strategies**   Sampling-based decoding algorithms such as top-$k$ [7] and top-$p$ (nucleus sampling) [13] are widely used in the various applications [1, 30] and large models [29, 5]. We also confirm that our DITTO is compatible with these popular stochastic decoding strategies. The results are presented in Table 2. It can be seen that models trained with DITTO can reduce repetitions compared to the MLE baseline and achieve the closest results to human in repetition metrics. For the quality of generated sequences, DITTO achieves the highest MAUVE of 0.96 with top-$k$ or nucleus sampling.

**Human Evaluation**   We conduct a pairwise crowdworker evaluation to judge the quality of the generations of DITTO compared to other baselines. For each pair of methods, the model generates texts based on the same 100 random prefixes from the test set of Wikitext-103. Evaluators are asked to independently judge which generation is better in terms of their grammaticality, relevance, coherence and informativeness. The evaluation interface and more details are in Appendix E. As shown in Table 3, DITTO consistently and significantly outperforms other competitors across different decoding strategies.

**Self-reinforcement Effect**   As mentioned in Sec. 2, the sentence-level repetition has a self-reinforcement effect, which is the core issue of sentence-level repetition. We further study it when the model is trained with different

Table 3: Human Evaluation Results. * means the results are statistically significant (2-sided binomial test, $p<.05$).

| | Win Rate |
|---|---|
| *Greedy Search* | |
| **DITTO** vs MLE | *84% |
| **DITTO** vs UL-token | *80% |
| **DITTO** vs UL-token+seq | *65% |
| **DITTO** vs SG | *68% |
| *Nucleus Sampling $p = 0.9$* | |
| **DITTO** vs MLE | *65% |
| **DITTO** vs UL-token | *71% |
| **DITTO** vs UL-token+seq | *62% |
| **DITTO** vs SG | *63% |

methods. The settings of experiments follow those in Sec. 2. As shown in Figure 4, we can find 1) TP,

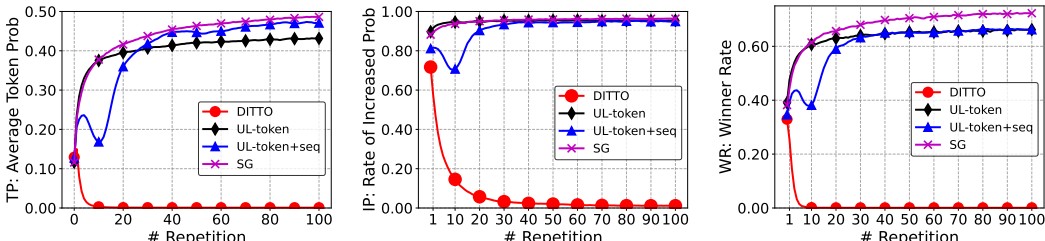

Figure 4: Results of different training-based methods by feeding repetitive sentences as described in Sec. 2]. We average the results on sentences from $\mathcal{D}_{\text{wiki}}$.

IP and WR increase almost monotonically in UL and SG, which shows that the self-reinforcement effect has not been solved in these training-based methods; 2) When the model is trained with DITTO, these metrics drop rapidly as the number of repetitions grows, showing the effectiveness of DITTO in overcoming the self-reinforcement effect.

## 4.3 Results of Directed Generation

Table 4: Abstractive summarization results on CNN/DailyMail.

| Model | ROUGE-1 | ROUGE-2 | ROUGE-L |
|---|---|---|---|
| Pointer-generator + Coverage [32] | 39.53 | 17.28 | 36.38 |
| Mask Attention Network [8] | 40.98 | 18.29 | 37.88 |
| BertSum [20] | 42.13 | 19.60 | 39.18 |
| UniLM [6] | 43.08 | 20.43 | 40.34 |
| UniLM V2 [2] | 43.16 | 20.42 | 40.14 |
| ERNIE-GEN-large [37] | 44.02 | 21.17 | 41.26 |
| PEGASUS [38] | 44.17 | 21.47 | 41.11 |
| ProphetNet [28] | 44.20 | 21.17 | 41.30 |
| PALM [4] | 44.30 | 21.12 | 41.14 |
| BART-large w.t. MLE [17] | 44.11±0.03 | 21.21±0.01 | 40.83±0.02 |
| BART-large w.t. UL-token [36] | 44.17±0.04 | 21.20±0.02 | 40.83±0.03 |
| BART-large w.t. UL-token+seq [36] | 44.13±0.07 | 21.15±0.11 | 40.71±0.09 |
| BART-large w.t. SG [19] | 44.18±0.06 | 21.17±0.07 | 40.89±0.05 |
| BART-large w.t. DITTO | **44.41±0.03** | **21.45±0.01** | **41.16±0.02** |

**Setup** We further conduct experiments on the directed abstractive summarization task CNN/DailyMail [12, 23]: given an input document, the model generates several sentences as summarization. We adopt the state-of-the-art model BART-large [17] as our baseline, which is large-scale encoder-decoder Transformer architecture trained on 160Gb data. For DITTO training, given a document and its summarization, we construct the pseudo data by first random sampling a sentence from the summarization and then repeating it until reaching the maximum summarization length of the decoder model while leaving the document unchanged. We follow the official implementations [17] to train the BART-large model on the CNN/DailyMail and then fine-tune the model with the best validation perplexity using DITTO training. The $\lambda$ is set as 0.9. During inference, tri-gram blocking [32, 23] and beam search (beam size = 5) are used as in Lewis *et al.* [17].

**Results** We evaluate the performance of the model with the standard F1-based ROUGE [31] scores (ROUGE-1, ROUGE-2, ROUGE-L). As shown in Figure 4, our training method consistently outperforms UL and SG with a large margin on ROUGE scores and outperforms other competitive baselines. The results also show that DITTO is compatible with the n-gram blocking, a decoding-based method for mitigating repetitions, demonstrating the generality of our approach.

## 4.4 Analyses

**Auto-completion with different decoding lengths** From a practice point of view, we analyze DITTO in different decoding lengths. As shown in Figure 5(a) and (b), the models trained with DITTO consistently and significantly outperform those trained with MLE with the constraints of different decoding lengths, which shows the effectiveness and robustness of our method.

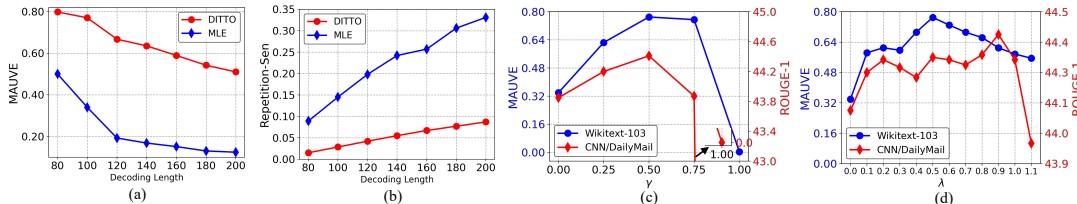

Figure 5: Results of DITTO in different decoding lengths and hyper-parameters. $\lambda$ is the penalization factor and $\gamma$ is the mix ratio of pseudo data and true data. Results are average of three runs with different random seeds.

**Hyper-parameter Study**   Towards better usage and understanding of DITTO, we study the two hyper-parameters: 1) mix ratio of pseudo data and actual data, denoted as $\gamma$, and 2) penalization factor $\lambda$. As is shown in Figure 5(c), models achieve the best performance with $\gamma = 0.5$ on both datasets. Thus, equally mixing the DITTO update and the MLE training loss update is recommended in practice. As for $\lambda$, we can observe that the optimal value varies in different tasks. In open-ended generation task Wikitext-103, a stronger penalization with $\lambda = 0.5$ is preferred. However, a mild penalization with $\lambda = 0.9$ is better in the summarization task. We conjecture that the repetition issue is more severe in generation tasks with more freedom, such as the open-ended generation.

## 5   Related Work

**Repetition in Neural Text Generation.**   Repetition has been a key problem in neural text generation for various tasks including open-ended generation (e.g., text continuation [13, 36, 19]) and direct generation (e.g., summarization [32, 20]). Previous work [13, 36, 18, 15] observes that, with maximization-based decoding strategies such as greedy search, language models prefer to generate bland and *consecutive* repetitions at the word-level, phrase-level, and sentence-level. Consecutive word-level and phrase-level repetition are lexical repetitions [10] where the sentence structure is incorrect, while sentence-level repetition is semantic repetition that indicates the expression is not informative and has a discrepancy with human language [10]. Recently, large-scale pre-training with Transformer architecture [34] such as GPT-2 [29] and BART [17], have greatly improved the ability of language modeling and can generate fluent sentences similar to human language. However, the generations still have a large number of unexpected consecutive sentence-level repetitions [29, 5, 9] in practice, which remains a severe problem in neural text generation.

He *et al.* [11] find that, when given the ground-truth context as the prefix or repetitive sentences generated by the model itself as the prefix, the language model can generate high quality texts which shows the model has the self-recovery ability. However, they adopt ancestral sampling to generate continuations rather than maximization-based decoding methods. When the model generates tokens by ancestral sampling such that the current sentence structure is different from previous sentences, the model stops repeating the sentence since their sentence-level context are different. Different from their findings, our analyses reveal that the probability of generation repetitive tokens will increase if they share the same sentence-level context.

**Approaches for Mitigating Repetition.**   Approaches for mitigating repetition in neural text generation can be categorized into *training-based* [36, 19, 33] and *decoding-based* [32, 7, 13, 33] approaches. The representative training-based approaches are unlikelihood training (UL) [36] and straight to gradient (SG) [19]. UL reduces the probability of negative candidate tokens, and SG improves the probability of tokens that do not belong to negative tokens, where the negative tokens are a set of tokens that appear in the previous context. However, our experiments in Sec. 4.2 show that they cannot solve *self-reinforcement* issue. In contrast, our work analyzes why models prefer to repeat themselves and quickly get stuck into the sentence-level loop, and mitigate the self-reinforcement issue. Many decoding-based methods have been proposed to rectify these issues in various tasks. In summarization, $n$-gram blocking [32, 23, 17] is often used to block repetitive $n$-grams from subsequent generations. In open-ended tasks, top-$k$ [7] and nucleus sampling [13, 35] truncate unreliable tail and employ sampling according to token distribution for generating fluent and informative texts. SimCTG [33] proposes contrastive search to reduce repetitions by first calibrating the model's representation space and then encouraging a trade-off between diversity and coherence in the generated text.

**Analyses of Repetition Problem.** Although previous work has noticed the repetition issue and many approaches have been proposed to mitigate it at the model training or decoding stages, little has been discussed about the causes of the repetition. Fu *et al.* [9] theoretically analyzes the problem by assuming the language models can be approximated to first-order Markov models. However, the cases of positive feedback loop of repetitions presented by Holtzman *et al.* [13] indicate the repetition probability has complex relationships with a quite long-range context and language models may not be simplified as first-order Markov models. However, there is a lack of detailed numeric and statistical results across various sentences and analyses in their work. In contrast, we further dig into the problem and conduct quantitative experiments to analyze underlying issues of repetition.

# 6 Conclusion and Future Work

We quantitatively investigate why language models with maximization-based decoding prefer consecutive sentence-level repetitions. Our observations and analyses show that the issue of the repetition is the tendency to repeat previous sentences and the self-reinforcement effect. Guided by our analyses, we propose a simple and effective method named DITTO by constructing pseudo data and teaching model to learn to gradually decay the repetition probability as the number of sentence repetitions grows. The experiments on the Wikitext-103 open-ended generation task and the CNN/DailyMail summarization task demonstrate the superiority of our methods.

This work investigates the relationship between the probability of the token and the number of repetitions in context. However, there should be deeper reasons for why the model raises the probability of repeating tokens from the perspective of model embedding, neural network architecture or intrinsic characteristics of language. Our current analysis has not touched these aspects, which we leave for future work. We hope our quantitative analyses and approaches to mitigate sentence-level repetitions can help the NLP community better understand, resolve the repetition issue and improve the fundamental language modeling ability of neural language models.

# 7 Acknowledgements

The authors thank three anonymous reviewers and meta reviewer for their comments, which greatly improved the article. Jin Xu and Jian Li are supported in part by the National Natural Science Foundation of China Grant 62161146004, Turing AI Institute of Nanjing and Xi'an Institute for Interdisciplinary Information Core Technology.

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
