# Appendix of 'Learning to Break the Loop: Analyzing and Mitigating Repetitions for Neural Text Generation'

**Jin Xu**[1]*, **Xiaojiang Liu**[4], **Jianhao Yan**[2], **Deng Cai**[3], **Huayang Li**[4], **Jian Li**[1]

[1]Institute for Interdisciplinary Information Sciences, Tsinghua University
[2]School of Engineering, Westlake University
[3]The Chinese University of Hong Kong
[4]Apple

xujin21@mails.tsinghua.edu.cn, xiaojiang_liu@apple.com, elliottyan37@gmail.com
thisisjcykcd@gmail.com, hli46@apple.com, lijian83@mail.tsinghua.edu.cn

## A  Consecutive Repetitions and Statistics of Beam Search Results

Previous work [2, 1] has observed that standard training and greedy decoding usually cause models to generate consecutive repetitive texts. These consecutive repetitive texts are redundant and do not convey new information, which is avoided in human language. There are three types of consecutive repetitions: word-level, phrase-level and sentence-level. The phrase-level means that a phrase consisting of several words is repeated consecutively. The sentence in our paper refers to a sequence split by '.!?' is repeated consecutively [2]. We calculate the ratio of consecutive repetition in a sequence $\mathbf{x}$ as follows.

**Consecutive word- and phrase-level repetition**  Denote a sequence as $\mathbf{x} = (x_1, \cdots, x_{|\mathbf{x}|})$. The word-level repetition is calculated by $\frac{1}{|\mathbf{x}|-1} \sum_{i=2}^{|\mathbf{x}|} \mathbb{1}(x_i = x_{i-1})$ where $\mathbb{1}$ refers to indicator function. The phrase-level repetition where the phrase has $k$ words is calculated by $\frac{1}{|\mathbf{x}|-2k+1} \sum_{i=2k}^{|\mathbf{x}|} \mathbb{1}((x_{i-k+1}, \cdots, x_i) = (x_{i-2k+1}, \cdots, x_{i-k}))$. We calculate them for each sequence $\mathbf{x}$ and average over the whole corpus.

**Consecutive sentence-level repetition**  Denote a sample sequence as $\mathbf{x} = (\mathbf{s}^0, \cdots, \mathbf{s}^N)$ that contains $(N+1)$ sentences. The sentence-level repetition is calculated by $\frac{1}{N} \sum_{i=1}^{N} \mathbb{1}(\mathbf{s}^i = \mathbf{s}^{i-1})$. We calculate it for each sequence $\mathbf{x}$ and average over the whole corpus.

As discussed in Section 1, compared to human language, the model with greedy decoding has substantially more sentence-level repetition. This phenomenon holds for other maximization-based decoding methods, such as beam search shown in Figure 1.

## B  Self-reinforcement in Model Generated Texts

When decoding auto-regressively, the probabilities of the repetitive sentence loops also have a self-reinforcement effect. As shown in Figure 2, the probability of the token 'located' increases almost monotonically with the number of historical repetitions, which shows the same trend as in Figure 1.

**Why manually repeat sentences to observe the probabilities rather than directly observe those of repetitive texts generated by the model auto-regressively**  Since our target is to study the

---

*The work was conducted in Apple.

[2]The strict definition of the sentence can be found at `https://en.wikipedia.org/wiki/Sentence_(linguistics)`. Here we use the end token to split sentences for ease of experiments.

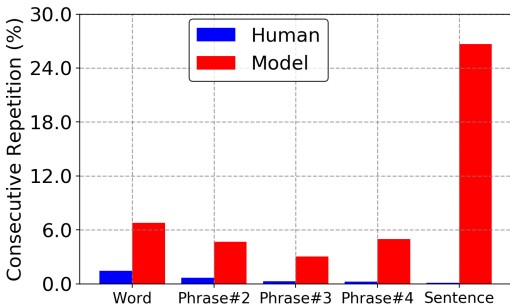

Figure 1: Statistics of human language versus model generation. The model generates the next 200 tokens with beam search ($b = 10$) given 50 tokens as the prefix. The results are the average of the Wikitext-103 dev set. Compared to human language, the model has substantially more consecutive *sentence-level* repetition.

**Context:** The rim is slightly raised about the surrounding surface and it has an outer rampart that has been only lightly impacted . No significant craters intersect the rim , and it is sloped about 1 @.@ 5 ° toward the direction 50 – 90 ° from the Earth .

**Greedy Decoding:**
The age of the crater is about 3 @.@ 6 billion years and it has been in the proximity of the south lunar pole for at least 10 @,@ 000 years . The South Crater is located on the southern edge of the northern highlands . The South Crater is located on the southern edge of the northern highlands . The South Crater is located on the southern edge of the northern highlands . The South Crater is located on the southern edge of the northern highlands . The South Crater is located on the southern edge of the northern highlands . The South Crater is located on the southern edge of the northern highlands . The South Crater is located on the southern edge of the northern highlands . The South Crater is located on the southern edge of the northern highlands . ...

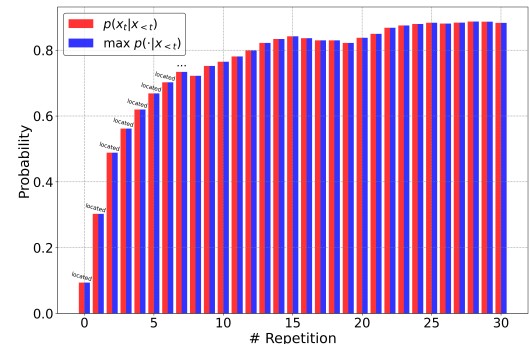

Figure 2: As mentioned in Figure 1, given the prefix, the model gets stuck in the sentence-level loop (i.e., repeat the sentence 'The South Crater is located on the southern edge of the northern highlands .'). We present the probability of the token 'located' (*y-axis*) as the number of historical repetitions (*x-axis*) grows. For texts generated by the model autoregressively, the probability of repetition (in red) also has a *self-reinforcement effect*. Best viewed in color and zoomed in a desktop monitor.

relationships between the probabilities of repetitive tokens with their repetitions in context, we can manually repeat the sentence and observe the probabilities of the next repetition to simulate the situation when decoding auto-regressively. We do not directly observe the probabilities of repetitions across various generations and calculate metrics $\text{TP}_n$, $\text{IP}_n$ and $\text{WR}_n$, as described in Section **??**, because 1) The model does not always get stuck into the repetitive sentence; 2) There are usually some 'dirty' tokens such as '' inserted into the consecutive sentences (e.g., the sentence in Table 3) or some minor changes (e.g., the sentence in Table 4)) in consecutive sentences repetitions so that it is not feasible to calculate those metrics for every token in sentences; 3) Most importantly, we want to study whether the self-reinforcement effect is caused by some problems in the language modeling ability itself so that the effects exists for not only the sentences generated by the model but also any sentences no matter whether the sentences repeated in contexts are grammatically correct.

## C  Loss Function of DITTO

The training loss function of DITTO is

$$\mathcal{L}_{\text{DITTO}}^{n,l}(\mathcal{P}_\theta(x_{n,l}|\mathbf{x}_{<n,l})) = -\log(1 - \left|\mathcal{P}_\theta(x_{n,l}|\mathbf{x}_{<n,l}) - \lambda \cdot \mathcal{P}_\theta^*(x_{n-1,l}|\mathbf{x}_{<n-1,l})\right|), \quad (1)$$

where $\mathcal{P}_\theta^*(\cdot)$ means that the value is excluded for gradient backpropgation and $\lambda$ is the penalization factor. The loss function is minimized when $\mathcal{P}_\theta(x_{n,l}|\mathbf{x}_{<n,l}) = \lambda \cdot \mathcal{P}_\theta^*(x_{n-1,l}|\mathbf{x}_{<n-1,l})$ where $\lambda$ is a key hyper-parameter. When $\lambda = 1$, the loss function requires the probability of token $x_{n,l}$ in $n$-th repetition is same as that in the $(n-1)$-th repetition to avoid the self-reinforcement effect; when $\lambda < 1$, the probability of token $x_{n,l}$ in $n$-th repetition should be smaller than that in the $(n-1)$-th repetition to make model averse to sentence-level repetition. To achieve this goal, we can use the following loss function:

$$\mathcal{L}_{\text{DITTO-mse}}^{n,l}(\mathcal{P}_\theta(x_{n,l}|\mathbf{x}_{<n,l})) = \text{MSE}(\mathcal{P}_\theta(x_{n,l}|\mathbf{x}_{<n,l}), \lambda \cdot \mathcal{P}_\theta^*(x_{n-1,l}|\mathbf{x}_{<n-1,l})), \quad (2)$$

where MSE refers to mean square error. However, in practice, we find that the MSE loss achieves inferior performance. The results are shown in Table 1.

Rather than requiring the probability of token $x_{n,l}$ in $n$-th repetition is the same as that in the last repetition multiplying $\lambda$, we can relax the constraint and enable the probability of token $x_{n,l}$ to be no more than that in the $(n-1)$-th repetition multiplying $\lambda$ as

$$\mathcal{L}_{\text{DITTO-margin}}^{n,l}(\mathcal{P}_\theta(x_{n,l}|\mathbf{x}_{<n,l})) = \begin{cases} 0, & \text{if } \mathcal{P}_\theta(x_{n,l}|\mathbf{x}_{<n,l}) \leq \lambda \cdot \mathcal{P}_\theta^*(x_{n-1,l}|\mathbf{x}_{<n-1,l}) \\ \mathcal{L}_{\text{DITTO}}^{n,l}, & \text{otherwise} \end{cases}. \quad (3)$$

In practice, as shown in Table 1, $\mathcal{L}_{\text{DITTO-margin}}$ achieve worse results. We further analyze its performance on the self-reinforcement effect. As shown in Figure 3, the model trained with $\mathcal{L}_{\text{DITTO-margin}}$ quickly reduces TP to close to 0 even if there is only one sentence repetition. It indicates that the model may have learned a 'cheap' solution to optimize the loss function $\mathcal{L}_{\text{DITTO-margin}}$: Regardless of the probability of the previous sentence $\mathcal{P}_\theta^*(x_{n-1,l}|\mathbf{x}_{<n-1,l})$, the probability of repeating the previous sentence is directly reduced to 0. Thus, the model may over-penalize repetitions so that all repetitions are forbidden in generations. However, there are some necessary repetitions that naturally occur, as discussed in Section 4. Over-penalizing repetitions may hurt the language modeling ability and thus lead to inferior performance.

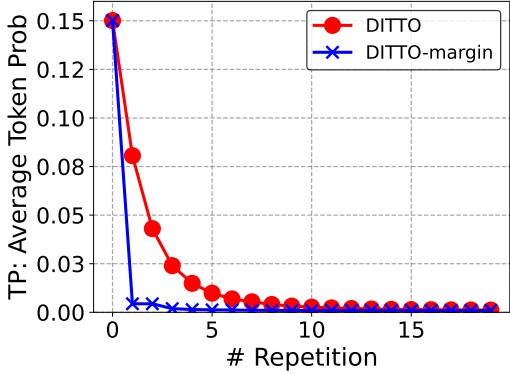

Figure 3: Comparison of sentence probability. The model trained with $\mathcal{L}_{\text{DITTO-margin}}$ quickly reduces SP to close to 0 even if there is only one sentence repetition.

Table 1: Results of different loss functions on the test subset of Wikitext-103 for the open-ended generation task.

| Objective | MAUVE | Perplexity | Accuracy | Repetition-4 | Repetition-Sen |
|---|---|---|---|---|---|
| $\mathcal{L}_{\text{DITTO-mse}}$ | $0.73_{\pm0.01}$ | $24.34_{\pm0.04}$ | $0.42_{\pm0.00}$ | $22.70_{\pm0.34}\%$ | $2.98_{\pm0.77}\%$ |
| $\mathcal{L}_{\text{DITTO-margin}}$ | $0.66_{\pm0.04}$ | $24.38_{\pm0.04}$ | $0.41_{\pm0.00}$ | $\mathbf{19.31_{\pm0.44}\%}$ | $\mathbf{2.17_{\pm0.77}\%}$ |
| $\mathcal{L}_{\text{DITTO}}$ | $\mathbf{0.77_{\pm0.01}}$ | $\mathbf{24.33_{\pm0.04}}$ | $\mathbf{0.42_{\pm0.00}}$ | $22.00_{\pm0.31}\%$ | $2.85_{\pm0.74}\%$ |
| Human | - | - | - | $1.10\%$ | $0.01\%$ |

# D   Additional Analyses and Experiments

## D.1   Self-reinforcement at sentences with different initial probabilities

To further verify that sentences with higher initial probabilities usually have a stronger self-reinforcement effect, we equally divide the $\mathcal{D}_{\text{wiki}}$ into 5 groups according to their initial probabilities $\text{TP}(\mathbf{s^0})$ and then repeat the experiments as described in Section 2. As shown in Figure 4, sentences with higher initial probabilities reach higher TP and WR as the number of repetitions increases, meaning that these sentences are more likely to be repeated (**stronger self-reinforcement effect**).

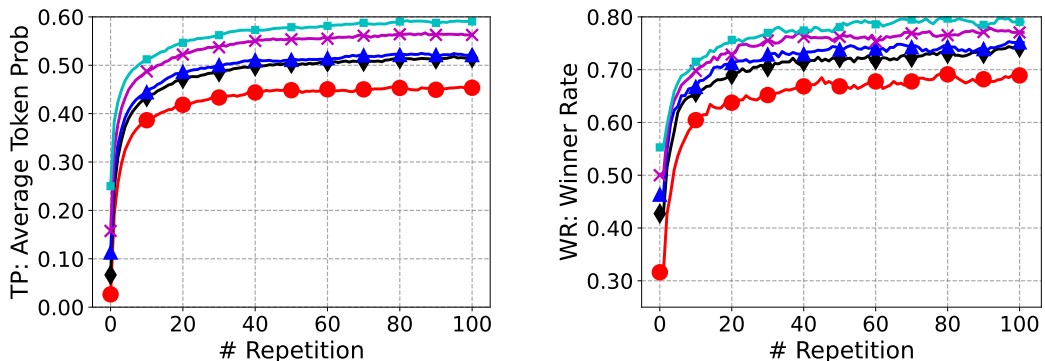

Figure 4: Results of TP and WR by feeding repetitive sentences as described in Sec.2. We equally divide the $\mathcal{D}_{\text{wiki}}$ into 5 groups according to their initial probabilities $\text{TP}(\mathbf{s^0})$ and then average the results.

Table 2: Results of DITTO with the group-level repetitive data. The results are reported based on three runs with different random seeds on the test set of Wikitext-103.

| Model | MAUVE | Perplexity | Accuracy | Repetition-4 | Repetition-Sen |
|---|---|---|---|---|---|
| DITTO w.t. Two Sentences | $0.77_{\pm 0.01}$ | $24.37_{\pm 0.03}$ | $0.42_{\pm 0.00}$ | $24.20_{\pm 0.38}\%$ | $2.99_{\pm 0.89}\%$ |
| DITTO | $0.77_{\pm 0.01}$ | $24.33_{\pm 0.04}$ | $0.42_{\pm 0.00}$ | $22.00_{\pm 0.31}\%$ | $2.85_{\pm 0.74}\%$ |
| Human | - | - | - | $1.10\%$ | $0.01\%$ |

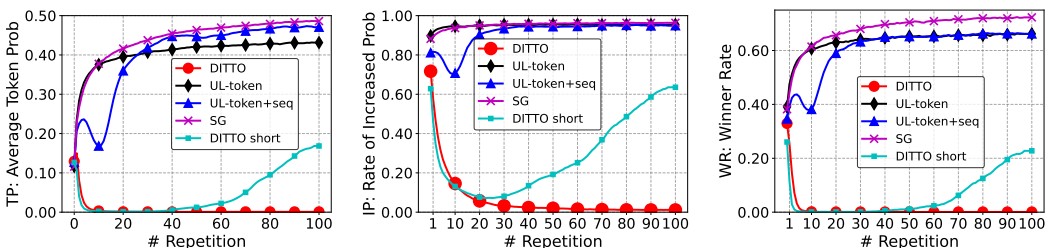

Figure 5: Results of different training-based methods by feeding repetitive sentences. 'DITTO short' means we only feed repetitive sentences with a maximum input length of 150 tokens for training. We average the results on sentences from $\mathcal{D}_{\text{wiki}}$.

For decoding scenarios, if maximization-based decoding algorithms such as greedy decoding are employed, previously generated sentences have a higher initial likelihood (since these sentences themselves are selected with the maximization criterion) and thus, have higher probabilities of being repeated.

## D.2 Study of Pseudo Repetitive Data

In this section, we study other different choices to construct pseudo repetitive data.

**Repeating Two Sentences as Pseudo Data**   Rather than repeating one sentence to construct pseudo data, we can randomly pick two consecutive sentences as a group and construct pseudo data by repeating the group. Then, we apply repetition penalization as Eqn. 2 at the group level. As shown in Table 2, repeating two sentences as pseudo data for training achieves similar results as the original one.

**Length of Pseudo Repetitive Data**   DITTO constructs pseudo repetitive data by repeating sentences until reaching the maximum input sequence length of the model (e.g., 1536 tokens). To further study whether the length of pseudo data has an effect on overcoming the self-reinforcement effect, we short the maximum input length of the repetitive sequence to 150 tokens for training, named DITTO-short.

Then, we measure the TP, IP, and WR metrics. The results are shown in Figure 5. From Figure 5, we can see that, for the short-decoding lengths (e.g., generating the next 100 tokens in open-ended generation tasks), DITTO-short can effectively reduce the values of TP, IP and WR. However, for long decoding length, TP, IP, and WR gradually increase at the end, which does not share the case with DITTO, showing that long sequence penalization is necessary to overcome the self-reinforcement effect. Compared with UL-token+seq [2], which also uses 150 tokens for penalization, DITTO enjoys two benefits: 1) DITTO can directly feed longer sequences for penalization training without significantly increasing the computational cost while UL-token+seq needs auto-regressive generation; 2) with the same penalization length (150 tokens), DITTO is more effective on overcoming self-reinforcement effect (e.g., lower TP, IP and WR values).

## E  Human Evaluation Details

We conduct a pairwise crowdworker evaluation to judge the quality of the generations of DITTO compared to other baselines. Models generate continuations based on the same 100 random prefixes from the test set of Wikitext-103. For each comparison of two continuations, three fluent English speakers are asked to independently judge which continuation is better. The overall quality is judged from four aspects: 1) grammar correctness; 2) relevance; 3) coherence; and 4) informativeness. The win rate is calculated as the total number of times that DITTO beats the other model divided by the total number of comparisons in the evaluation. The interface for the human evaluation is shown in Figure 6.

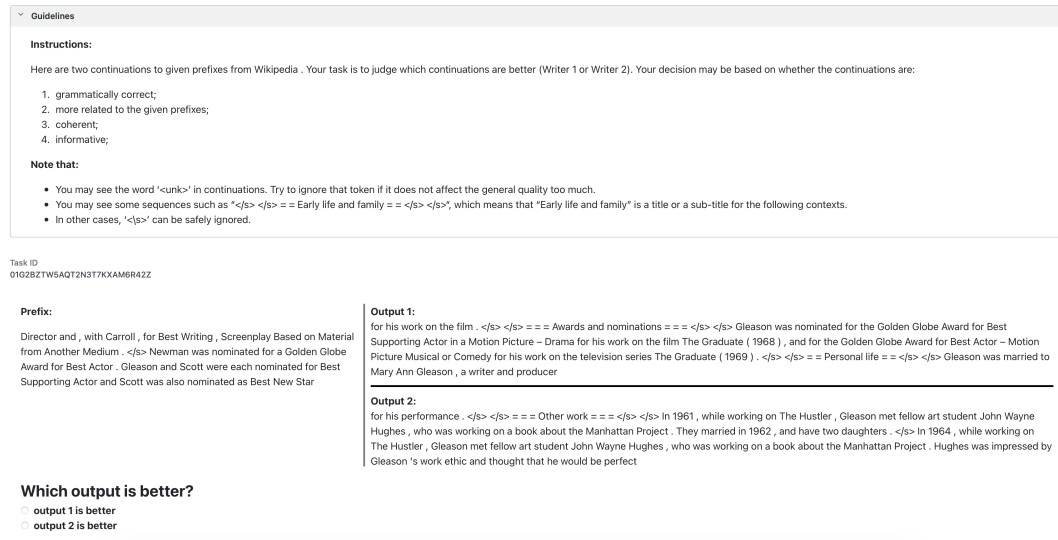

Figure 6: Screenshot of the user interface in human evaluation experiments.

## F  Examples

In this section, we show the examples of generated texts of different training methods in the open-ended generation task. The examples are presented in the following tables.

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

Table 3: Example 1 of generated texts by models with greedy decoding.

**Prefix**

was an immediate public outcry following the publication of the reports , particularly with regards to the decision to not repatriate the bodies of the dead . The reports generated considerable discussion in the press which ultimately led to a heated debate in Parliament on 4 May 1920 . Sir

**MLE**

Edward Cave , the Director of the Public Prosecutions , was quoted as saying that Cave had " no idea what the truth was , and that the truth was not the truth. But what the truth was , and that the truth was not the truth . But what the truth was , and that the truth was not the truth . But what the truth was , and that the truth was not the truth .  But what the truth was , and that the truth was not the truth . But what the

**UL-token**

John Russell , who had been appointed High Commissioner for Ireland , was also present at the hearing .   = = = = Public response = = = =   The public response to the report was mixed . The Irish Times called it " a most damning and comprehensive report on the Irish Question " . The Irish Times called it " a book of great importance " and " a work of great importance " . The Irish Times called it " a work of great importance " and " a work of great importance

**UL-token+seq**

Geoffrey Howe , the Prime Minister , and his wife were among the signatories to the London Necropolis Act , which was passed on 30 June 1921 .   = = = = The London Necropolis Railway Act 1921 = 6 1 / 2-mile ( 8.0 km ) long railway from Brookwood to Brookwood was the first railway in the world to be built in the United States outside the British Empire . The Act allowed the LSWR to carry out the work by rail from its own station at Brookwood , and the LSWR

**SG**

Edward Cave read the report and stated that he had been " deeply disappointed " that the government had not acted on the grounds that it had not acted on the grounds that it had not acted on the grounds that it had not acted on the grounds that it had not acted on the grounds that it had not acted on the grounds that it had not acted on the grounds that it had not acted on the grounds that it had not acted on the grounds that it had not

**DITTO**

Edward Cave , the Director of Public Prosecutions , was quoted as saying that " the Government has not been able to make a decision on the issue of the bodies of the dead . "   = = = Public reaction = = =   The public reaction to the killings was generally positive . The Times called the actions " a most appalling and appalling act " and the Daily Mail called for the police to be withdrawn . The Daily Mail called for the police to be withdrawn , but the Daily Mail said that

Table 4: Example 2 of generated texts by models with greedy decoding.

**Prefix**

is summed up as " not to be missed . "  Paul Newman reprised his role as Fast Eddie Felson in the 1986 film The Color of Money , for which he won the Academy Award for Best Actor in a Leading Role . A number of observers and

**MLE**

critics have praised Fast Eddie Felson as one of the greatest actors of all time . In 2007 , Fast Eddie Felson was ranked # 1 on Empire magazine 's list of the 500 Greatest Movie Characters of All Time . In 2010 , Fast Eddie 's popularity was listed at # 3 on Empire 's list of the 500 Greatest Movie Characters of All Time . In 2012 , Fast Eddie 's popularity was ranked # 1 on Empire 's list of the 500 Greatest Movie Characters of All Time . In 2013 , Fast Eddie 's popularity was

**UL-token**

critics praised his performance . The New York Times ' Janet Maslin wrote , " Fast Eddie Felson is a real joy to watch , and he is a real joy to watch . " The New York Times ' Janet Maslin wrote , " Fast Eddie Felson is a real joy to watch , and he is a real joy to watch . " The New York Times ' Janet Maslin wrote , " Fast Eddie Felson is a real joy to watch , and he is a real joy to watch . " The New York Times

**UL-token+seq**

critics have praised Fast Eddie Felson as the film 's most memorable moment . In his review for the Chicago Reader , critic Richard Schickel wrote that " Fast Eddie Felson is the first of the great actors to come across as a master of the art of acting . " In his review for the New York Times , critic Anthony Boucher wrote that " Fast Eddie Felson is the first actor who can make a living through the motions of his actors , and the film is a triumph of invention . " In his review for the

**SG**

critics have praised Fast Eddie 's performance , including Roger Ebert , who wrote that " Fast Eddie is a great comic actor , and he has a great comic timing . "   = = Personal life = =   Fast Eddie was married twice , first to Patricia ( née <unk> ) and then to Patricia ( née <unk> ) . They divorced in 1977 . Their son , Eddie Jr . , is a retired basketball player who played for the Denver Nuggets , New York Knicks , Boston Celtics , and Dallas Mavericks .

**DITTO**

critics have praised Fast Eddie 's performance , including the New York Times critic Bosley Crowther , who wrote that " Fast Eddie is a fine actor , and he is a fine actor . "   = = = Accolades = = =   Fast Eddie was nominated for the Academy Award for Best Actor in a Leading Role for his role as the title character in the 1986 film Fast Eddie . He was also nominated for the Golden Globe Award for Best Actor – Motion Picture Musical or Comedy for his role as the title

Table 5: Example 1 of generated texts by models with nucleus sampling.

| **Prefix** |
| --- |
| 's screenplay was selected by the Writers Guild of America in 2006 as the 96th best motion picture screenplay of all time . In June 2008 , AFI released its " Ten top Ten " — the best ten films in ten " classic " American film genres — after |
| **MLE** |
| a deal with Chase & Caldera Entertainment was reached . Ten countries also released 100 top ten films in multiple-day international markets .   = = = 2010s = = =    = = = = 2010s = = = =    = = = = = 2010s = = = = =    = = = = = = 3D movies = = = = = =    = = = = 3D movies |
| **UL-token** |
| adjusting their publication guidelines for rated films and having it between 25th and 26th , with cumulative scores of 70.2 and 90.5 . In December 2011 , The Hollywood Reporter said that with those achieved in reverse , characters such as Hopper could become " more powerful than previously unseen " . It also said that women would have to look after the characters in pictures like Barbie and Paris Hilton for inspiration .  In April 2014 , Warner Bros. was sued by Roger Hathaway , David Fincher , Arnold Schwarzenegger , Darren Aronofsky , |
| **UL-token+seq** |
| polling for AFI 's 10 Top 10 , and also including WALL-E , WALL Scattered , WALL Building 3 , and WALL Jack .  In October 2013 , Netflix announced the long-running website Expedition to Antarctica , celebrating the new Space Travel , and the search for and rescue mission planners had been in the middle of the year . A 2012 study by video game journalist Toby Philpott found a large majority of the space explorers who have survived on the ship as well as exploring and exploring locations . The Discovery and Curiosity |
| **SG** |
| polling over 300 people from 108 countries worldwide . In 2012 , AFI ranked 65th on Bravo 's 100 Years ... 100 Movies list .   = = Awards = =   Writers Guild of America 's 50 Years ... 100 Heroes & Villains :  ⚮ AFI 's 100 Years ... 100 Movie Quotes : " Run to your sister , son ... If you hate her [ ... ] you wouldn 't believe that you 're too stupid ? " – Nominated  AFI 's 100 Years ... 100 Thrills – Nominated  AFI 's |
| **DITTO** |
| polling more than 130,000 people at the screening and success .  In October 2009 , AFI 's 10 Top 10 ranked " The 100 Scariest Movie Trailer " , the first-ever list of AFI 's 100 Years ... 100 Thrills — while the film was also ranked among the top ten all-time greatest movie villains in the last five years — and in December 2012 , Time magazine listed it as the sixth-greatest film villain in history . In August 2012 , Total Film ranked " The 100 Scariest Movie Characters |

Table 6: Example 2 of generated texts by models with nucleus sampling.

**Prefix**

business activities upon his return . He invested in a wide variety of businesses and infrastructure , spurring economic activity in the state . His investments ranged widely , including maritime insurance ( where is father-in-law had made his fortune ) , bridges , locks ,

**MLE**

power plants and homes . He and his wife died from injuries sustained in labor ventures when they were old . He was buried in Père Lachaise Cemetery in 1919 . He left a lasting legacy to his family .  After leaving politics , Holloway became close friends with Josiah Quincy III .   = = Early life and family = =   Holloway was born on 12 November 1827 in Saint Joseph , Missouri , the only child of Elias and Emily Holloway . Most of Holloway 's family was derived from him , though they

**UL-token**

dams , gambling equipment , and positions .  Henderson lost heavily during his sojourn to Mississippi . He attempted to build a canal , which he provided on several occasions with investment in agriculture and other pursuits . His wife Grace later wrote that Henderson " loathed working men so long that it nearly destroyed the whole business " . She only learned to gamble at all levels of the United States in the late 1840s . When he had returned to Columbus in the mid-1830s , all those who had survived the Ohio River had little

**UL-token+seq**

a gas leak , and the offices of the administration , engineers , and messengers for scientific and industrial interests . He also invested in William <unk> Ingram who purchased two berths from the Sons of the American Revolution , a natural gas ship that had been fired in 1807 in retaliation for the order to resupply the British Army troops at Fredericksburg . He did not work on the reorganization of the army , but supported the need for self-government-managers in a transition known as Federal service .  Small farmers raised cattle in their

**SG**

tunnels , pumps , fire stations , telephones , showers and ships .  Retaining custody after the September 11 attacks , Romney returned to family life by appointing first-class city councilman Carl McCall as U.S. Attorney General . Later that year , he was recognized by the Sons of Prosperity organization , where he served as chair of the Drafting Committee on Government Reform . While in the mid-1980s , Romney unsuccessfully ran for re-election as Secretary of Commerce under Governor Lawrence B. Jennings , losing to Richards . While in office ,

**DITTO**

hotels , art galleries , and utilities . He formed new , larger enterprises ( offering bonds and furniture ) , offering services to smaller companies , attracted much investment in newly discovered companies . He also served as the state 's commissioner of the Board of Higher Education , providing education and higher education .   = = Personal life = =   John Smith was born in 1830 in Salt Lake City . His father , also named John , was an army officer stationed in a successful military career in the early 19th century .