# OpenReview forum: "Learning to Break the Loop: Analyzing and Mitigating Repetitions for Neural Text Generation "
_NeurIPS.cc/2022/Conference — NeurIPS 2022 Accept_

### Official Review · Reviewer_tE1F · 2022-07-05

**Rating:** 6
**Confidence:** 4
**Soundness:** 3 good
**Presentation:** 3 good
**Contribution:** 3 good

**Summary:**

This paper presents an empirical study showing the problem of repetitions in text generation (especially sentence repetitions), and proposed a training method called DITTO to address this problem where they add a training objective to penalize the probabilities of sentence level repetitions for the synthetically created examples (repeated sentences). Authors evaluated their method on different tasks and datasets, open domain text generation and summarization, and showed their method has much less repetition while still achieves competitive performance on other metrics.


**Questions:**

In sec 3:
Authors mentioned "We find that, appending another sentence sampled from the training corpus randomly as the prefix to the pseudo repetitive data x can achieve better and more stable performance."
Can you clarify this?  is this providing some context to the repetitive sentences? but if so, why do you need to randomly sample another sentence, rather than just the previous sentence?

Fig 4:  I’m a bit confused by the results for UL.
Earlier tables showed it has much lower repetition rate than other methods, which seems to be inconsistent with the results presented here.




**Limitations:**

Authors didn't discuss this.

**Strengths And Weaknesses:**

Strength:
The paper presents a good empirical analysis for the repetition problem in text generation.  Authors defined a few metrics, and evaluated on different datasets demonstrating the problem.  Though the community is aware of such problems, this is the first time I see such an analysis systematically showing the empirical results.

Based on the repetition pattens authors identified (higher probabilities when repetitive sentences are included in the context), authors proposed a training method to penalize such repetition. The proposed training objective is well motivated.


Weakness:

For the repetition problem, though the empirical analysis is good, I feel the authors are overclaiming slightly. They do show the sentence repetition happens, and models has a self-reinforcement effect. However, they say that’s the reason that sentence repetition occur (I.e., the model has seen the previous sentence in the context). I feel this is not exactly the explanation yet. The underlying cause of this phenomenon still needs additional analysis, probably more inside the neural networks.

---

> ### Author Response · Authors · 2022-08-02
> **Response to Reviewer tE1F**
>
> 1 **[Exact explanation about why sentence repetition occurs and discuss limitations]**
>
> We thank the reviewer for the insightful comments. In this paper, we explain why sentence repetition occurs by investigating the relationship between the probability of the token and the number of repetitions in context. However, we admit that there must be deeper reasons for why the model raises the probability of repeating tokens from the perspective of model embedding or neural network architecture. We refer the reviewer to the general response for a more detailed discussion. To avoid over-claiming, we will also discuss it in Sec.6.
>
> 2 **[Appending the previous sentence to the repetitive sentences]**
>
> We conduct experiments by appending the previous context to the repetitive sentences, as shown in Appendix D.2 and Table 3. From the table, we find that appending context as the prefix is slightly better in terms of repetition metrics. We will adopt it as the default method to construct repetitive sentences in the revised paper. Thanks for your advice.
>
> 3 **[Relationship between Table 1 and Figure 4]**
>
> Table 1 presents the results of generating the next 100 tokens. The average length of sentences is 11.4, so the results of 100 tokens correspond to the results of TP, IP, and WR of 8.77 repetitive sentences. For 8.77 repetitive sentences, TP(UL-token+seq) < TP(UL-token) $\approx$ TP(SG), which is consistent with the ranking of repetition metrics in Table 1. However, we need to point out that the results of Table 1 and Figure 4 are from two different settings, where the former is obtained with auto-regressive decoding and the latter is obtained with teacher-forcing decoding by feeding pseudo repetitive sentences. Thus, their rankings may not be strictly the same.

---

### Official Review · Reviewer_HDLP · 2022-07-08

**Rating:** 7
**Confidence:** 4
**Soundness:** 2 fair
**Presentation:** 3 good
**Contribution:** 3 good

**Summary:**

This work proposes a novel way of mitigating sentence-level repetition in autoregressive sequence generation: they form an artificial set of negative samples by repeating the same sentence from training corpus up to the max length of prediction. After that they use a loss function to penalize those instances from being produced during fine-tuning. They compared their method with multiple variants from previous work and concluded the superior quality of generated sequences. They have done both automatic and human evaluation for open-ended generation and automatic evaluation for directed generation.

**Questions:**

Here I will write both my comments and questions:

Line 30: [30] is also training based method.

Line 40-41: How does your analysis differ from one in Holtzman et al. work? I can see the introduction of several metrics to measure the rate of self-reinforcing loop, but other than that it doesn't look different. If you want to stress the point about analyzing sentence-level repeat, then it is a good place to do so.

Line 52-56: When you say "The cause ... may be that ...", then the overall claim does not sound strong at all. If you do not have sufficient evidence (e.g. by showing how this effect does not hold on some other test sets without repeats or so, then I am not sure how this is different from all other existing observations about n-gram level repeat.

Line 61, 63, 64: Please avoid using "can" or "may". if it is not always the case, then quantify how often this happens. After rewriting these sentences the overall reading will be improved.

Line 63-65: Where is the justification for this statement? It sounds incomplete without any reasoning afterwards.

Line 89: what does it mean to "statistically investigate"? Other wording will improve this sentence.

Line 123: "To our surprise": why is it surprising? All previous work was showing that this is indeed a large problem. I agree that random token sentence repeat is more surprising, but authors don't say that.

Lines 125-132: I do not see the answer to the question "Why sentence repetitions occur?". I see observations, but no verified hypothesis of the reason behind it. For instance, Contrastive Generation framework paper [30] specifically introduce the hypothesis of very similar embeddings of repeating tokens which they further verified. I do not see any similar process here. Perhaps authors had some valid answer, but it is not presented here.

Lines 133-137: Where is the answer for the question? The observation of self-reinforcing loop on artificial corpus is not an answer. Moreover, there is previous work showing that LM has the ability to break such a loop: https://arxiv.org/pdf/1905.10617.pdf I would be happy to see if authors can elaborate more about this.

Lines 144-153: Due to the unclear claims in the section 2.2 the statements in "Analyses" paragraph do not sound strong.

Line 167: why did you choose the loss function to look like this? It is interesting to see if there is some theoretical benefit of using such construction of log(1-|p1-p2|). For instance, the gradient of the loss may show some benefits. Alternatively you could have optimized the difference between log-probabilities: max(margin, logp(x_n) - logp(x_{n-1})) . Some justification around the loss function design would help readers to understand the motivation better.

Lines 190-193: At least for UL-seq it is not true that it penalize any repetitions. In fact it is specified by the length of ngram. In other words, if n=15, then it will only penalize those sentence-a-like ngrams.

Lines 256-258: DITTO is specifically designed to mitigate repeats up to max generation lengths, while other methods you compared to did not. In other words, other methods could have a chance to show better performance if they match the design of negative samples similarly to DITTO. And the same holds in opposite direction I believe: if DITTO would be trained for e.g. 150 tokens only, then it may fall in the same repetition loop again after generation that number of tokens. I think this is very important point to mention in this section.

Lines 305-207: As I said earlier, this can be a result of not 100% fair comparison between training and evaluation protocols.

**Limitations:**

Relevant related work which is missing (sorry if it is there and I missed it):

https://arxiv.org/pdf/1905.10617.pdf

https://arxiv.org/abs/2007.14966

https://www.semanticscholar.org/paper/High-probability-or-low-information-The-paradox-in-Meister-Wiher/49a7515b1c0250a12c0e6b6fa0a8b6d46fc42adf

https://arxiv.org/abs/2002.02492

**Strengths And Weaknesses:**

# Originality

## Strengths

* Authors proposed a novel way of viewing inter-sentence repetition which is a more global way compared to n-gram repetition (which can also be inter-sentence for large enough n-gram length)

## Weaknesses

* Many claims around idea justification are written in a vague way, which I will touch later in the questions section.

# Quality

## Strengths

* Authors uploaded a well-done code base with experiments and instruction of how to run it. The code is based on fairseq which makes it not hard to use in the future work by other researchers.

* Authors composed comparison with other methods which avoids this type of degeneracy in addition to the comparison with the baseline method.

## Weaknesses

* Related work section could be improved. For instance, there is a recent work (not related with myself in any way) about the ability of the LM to break the reinforcing loop: https://arxiv.org/pdf/1905.10617.pdf . In other words, there exists previous work directly related and opposite to observations and claims for this work. I want to stress that this is totally fine, but authors should consider discussing such results and providing some explanation of how the setup is different etc.

* Experiment comparison is not very transparent. For instance, sequence-level UL models are trained to penalize continuations generated by up to 150 tokens. It was observed earlier that after such 150 tokens it may tend to repeat again (looks like this is exactly what happens in Fig.4). In other words, the length of generation influences the ability of the model to not repeat up to such extent. In DITTO the model is being penalized by the max length of generation. You may have done the experiment where DITTO would be trained on negative sentences of similar length to match the UL training protocol. If you are not going to do that, then at least clarifying this is necessary, because it addresses your claim that UL does not break the reinforcing loop. Because it may do so if you train UL model in similar fashion (with longer generated seqs).

# Clarity

## Strengths

* The experimental protocol is clear.

## Weaknesses

* The section I was very curious about 2.2 is extremely vague and does not provide any answers (I will touch this later in the questions)

# Significance

## Strengths

* I believe that this general method provides a significant contribution for future work beyond this specific use case: using an external set of negative samples which are easy to form and optimize.

## Weaknesses

* On the other hand from the the strong point #1, the loss function is not well justified (I touch this again in the questions) as in why such construction (with modulo difference) is better than eg the margin based version.

* Even though Contrastive framework for text generation was discussed (it was mistakenly categorized as decoding based method while in fact it is training based too), it was not used for experimental comparison.

---

> ### Author Response · Authors · 2022-08-02
> **Response to Reviewer HDLP - Part 1/2**
>
> 1 **[Related Work]**
>
> He et al.[1] find that, when given the ground-truth context as the prefix or repetitive sentences generated by the model itself as the prefix, the LM can generate continuations with similar quality which shows the model has the self-recovery ability.
> However, during decoding, they adopt ancestral sampling to generate continuations rather than maximization-based decoding methods. When the model generates tokens by ancestral sampling such that the current sentence structure is different from previous sentences, the model stops repeating the sentence since their sentence-level context are different. Different from their findings, our analyses reveal that the probability of generation repetitive tokens will increase if they share the same sentence-level context. Thanks for the suggestion. We will cite the work and other relevant works mentioned in the comments, and discuss them in the related work section of the revised version.
>
> [1] Tianxing He, Jingzhao Zhang, Zhiming Zhou, James R. Glass. Exposure Bias versus Self-Recovery: Are Distortions Really Incremental for Autoregressive Text Generation? EMNLP 2021.
>
> 2 **[Comparison with UL-seq with short input pseudo data]**
>
> UL (Wellneck, et al., 2019) is the first work in mitigating repetition during the training phase, and our work is also motivated by the work. In our experiments, we exactly follow the original implementations of UL-seq. Following your advice, we re-train the DITTO by shorting the repetitive sequence to 150 tokens and study whether long sequence penalization is necessary to overcome the self-reinforcement effect. The results are shown in Appendix D.2 and Figure 10. From the results, we can find that long sequence penalization is actually necessary. However, compared with UL-token+seq, DITTO still enjoys two benefits: 1) DITTO can directly feed longer sequences for penalization training without significantly increasing the computational cost while UL-token+seq needs to auto-regressively generate sequences for penalization; 2) with similar penalization length (150 tokens), DITTO is more effective on overcoming self-reinforcement effect. We will add the discussion on the experimental parts in the revised paper. Thanks for your constructive suggestion!
>
> 3 **[Loss Function]**
>
> We thank the reviewer for the suggestion of modifying the loss function. We refer the reviewer to the general response above, where we argue why our loss function is more suitable than other losses such as the margin loss.
>
> 4 **[SimCTG is a training-based method. Comparison with SimCTG.]**
>
> Thanks for your advice. SimCTG is a contrastive framework to mitigate repetitions from both training and decoding. We have modified the description of the SimCTG. To further resolve your concern, we conduct experiments to compare DITTO with SimCTG. Note that the results of SimCTG in their paper are not comparable since they use bpe-level tokens and implement methods on the huggingface codebase. We reproduce their methods based on their public codes on the fairseq codebase and run the experiments at the word level. The results are shown in Appendix D.3 and Table 4. In most decoding algorithms including greedy decoding, top-k and nucleus sampling, DITTO can achieve
> superior performance. Especially, for top-k and nucleus decoding algorithms, DITTO can achieve the highest MAUVE score of 0.96 with higher accuracy and lower perplexity, demonstrating the effectiveness of our method.
>
> 5 **[Line 40-41: Difference with Holtzman et al. work]**
>
> Holtzman et al. present cases where the probability of a repeated phrase increases with each repetition. Since we find the model has a stronger preference for consecutive sentence-level repetitions, this paper mainly focuses on the sentence repetition issue. Compared to their work, our analyses have several differences: 1) we conduct quantitative experiments
> to show that self-reinforcement holds across various sentences; 2) Since we compare the self-reinforcement in various sentences, we find that sentences with higher initial probabilities have stronger self-reinforcement effect; 3) combined with maximization-based decoding algorithms, we explain why sentence-level repetition occurs (see also the general response to all reviewers). In sum, we provide a more systematic and quantitative study of the repetition problem and provide an explanation of repetition in the decoding phase base on our analyses.
>
> 6 **[Line 52-56: evidence of "The cause ... may be that ...", Line 61, 63, 64: using 'can' and 'may']**
>
> Thanks for your advice. We will re-phrase these sentences in the revised version.

---

> > ### Author Response · Authors · 2022-08-02
> > **Response to Reviewer HDLP - Part 2/2**
> >
> > 7 **[Line 63-65: justification of the statement]**
> >
> > To show that sentences with higher initial probabilities usually have a stronger self-reinforcement effect, we further conduct experiments to divide the $D_{wiki}$ into 5 groups according to their initial probabilities and compare their results as shown in Appendix D.1 and Figure 9. From the figure, it can be clearly seen that sentences with higher initial probabilities reach higher TP and WR as the number of repetitions increases, meaning that these sentences have higher probabilities of being repetitively generated (stronger self-reinforcement effect). For decoding scenarios, if maximization-based decoding algorithms such as greedy decoding are employed, previously generated sentences have a higher initial likelihood (since these sentences themselves are selected with the maximization criterion) and thus, have higher probabilities of being repeated.
> >
> > 8 **[Line 89: reword 'statistically investigate', Line 123: reword 'surprise']**
> >
> > Thanks for your comments. We have modified the word 'statistically' to 'quantitatively' and removed the phrase 'To our surprise' for a more precise expression.
> >
> > 9 **[Lines 125-132: answer to the question 'Why sentence repetitions occur?', Lines 133-137 answer to 'Why model get stuck into sentence-level loop'?, Lines 144-153, problems about analyses]**
> >
> > In this paper, we mainly explain why sentence repetition occurs from the probabilities of repeating previous sentences. However, we do not provide experiments to explain why the model raises the probability of repeating from the perspective of model embedding or architecture. We refer the reviewer to the general response for a detailed discussion about it and why the model gets stuck in the sentence-level loop. We believe that understanding why the model raises probabilities for repetition from the model and architecture perspective is an important research topic, which we are currently working on but is out of the scope of this paper. To make our claim clear to readers, we will add more discussion in Sec.6 about the limitation of our current analysis and future important directions.
> >
> >
> > 10 **[Line 190-193: UL-seq it is not true that it penalizes any repetitions]**
> >
> > We have modified the 'unlikelihood' to the 'token-level unlikelihood training'. Thanks!

---

> > > ### Comment · Reviewer_HDLP · 2022-08-03
> > > **Thanks for the details response!**
> > >
> > > Reviewers have addressed all questions I had. I have increased the rating.

---

### Official Review · Reviewer_95eQ · 2022-07-10

**Rating:** 6
**Confidence:** 3
**Soundness:** 3 good
**Presentation:** 3 good
**Contribution:** 3 good

**Summary:**

The paper investigates the repetition issue in natural text generation. The described loop issue was caused by the maximization-based decoding algorithm we use. Machine generated text has higher chance to be repetitive compared to human corpus. The paper starts with analysis on individual cases and then some more scalable experiments with a few standard metrics. It introduces a “self-reinforcement effect” where the model will be more likely to repeat if the similar context has shown before. The more times of previous occurrences, the more likely it will repeat. To tackle the problem, the paper proposes a simple and intuitive training objective, DITTO, to reduce the repetition. The paper conducts experiment on open ended text generation and text summarization on CNNDM.


**Questions:**

The equation 1 is designed to penalize “one-hop” similarity. Let’s say you have a sequence: The orange is blue. The orange is red. The orange is blue. The orange is red. [repetition]
In this case, every two sentences are different, but 2-sentence repetition is not a problem here. Is it possible to boost the equation so you can cover all the previous sentences?

**Limitations:**

The DITTO equation tackles limited scenario of repetition. Some data augmentation or contrastive learning techniques can be applied to make the training less “synthetic”.


**Strengths And Weaknesses:**

The paper tackles a core challenge in NLG. The “loop” of the paper is complete and convincing. The proposed technique is inspired by experiments and analysis. The idea of the paper is generally clear and easy to follow.
The proposed technique is relatively simple to implement and use.

It compares with some prior work and shows its effectiveness. Human evaluation is also provided to validate the proposed technique.

The empirical results are good. The proposed method is competitive compared to prior work including unlikelihood training and StraightGradient.

Weakness

I have a major question regarding the function of DITTO (Equation 1).
Equation 1:
-math.log( 1 - abs(cur - alpha * prev))
Let’s set alpha = 1 for now.
If the model is well-trained to avoid repetition, the current probability for a target token v P(x_{n,l} = v) is lower than the probability if shows before P(x_{n-1,l} = v), then this is an ideal scenario that the model did great. However, the equation actually penalizes this situation.
Here is my experiment:
(Good case) If cur=0.1, prev=0.5, alpha=1, the loss is 0.51.
 cur=0.1;prev=0.5;alpha=1
-math.log( 1 - abs(cur - alpha * prev ))
0.5108256237659907

(Bad case) If cur=0.9, prev=0.5, alpha=1, the loss is still 0.51
cur=0.9;prev=0.5;alpha=1
-math.log( 1 - abs(cur - alpha * prev ))
0.5108256237659907

This doesn’t make sense to me that the loss is the same for both cases. In the “good” case, the model decrease the chance of repetition so it should be rewarded rather than penalized.

If the alpha is set to be <1, it’s better than the case described when alpha=1, but still I think it’s unfair to penalize a model to not be repetitive.
cur=0.1;prev=0.5;alpha=0.5
-math.log( 1 - abs(cur - alpha * prev ))
0.16251892949777494
cur=0.9;prev=0.5;alpha=0.5
-math.log( 1 - abs(cur - alpha * prev ))
1.0498221244986778


If I understand the equation correctly, should it be max(_, 0) instead of abs(_)?

cur=0.1;prev=0.5;alpha=1
-math.log( 1 - max(cur - alpha * prev,0 ))
-0.0

cur=0.9;prev=0.5;alpha=1
-math.log( 1 - max(cur - alpha * prev,0 ))
0.5108256237659907
To conclude, the designed equation penalizes the model even the current pred has less “overlap” with previous pred. I am not sure if there is some misunderstanding of the equation. I apologize if there is, and I am open to discussion. Authors and reviewers are welcomed to comment on this.


The coverage of the experiment could be further improved. There are only two datasets used in this paper, Wiki103 for LM and CNNDM for summarization. It would be better if the paper can provide more evidence on more datasets.

---

> ### Author Response · Authors · 2022-08-02
> **Response to Reviewer 95eQ**
>
> 1 **[Design of the loss function]**
>
> We thank the reviewer for the suggestion of modifying the loss function. We refer the reviewer to the general response above, where we argue why our loss function is more suitable than other losses such as the margin loss.
>
> 2 **[Coverage of the experiments]**
>
> We thank the reviewer for the suggestion. In this paper, we follow most relevant previous works such as unlikelihood training (Wellneck et al., 2019) and straight to gradient (Lin et al., 2021) to conduct experiments on wikitext-103 and CNN/DM for comparison, and perform detailed studies on hyper-parameters and different decoding lengths. Due to the limited time for rebuttal, we leave more experiments on other tasks as future work.
>
> 3 **[Two-hop sentence repetition as pseudo data and more data augmentation technique for constructing pseudo data]**
>
>  In principle, we can view the two sentences as a group and then construct pseudo data by repeating the group for training. According to the advice, we conduct experiments and present the results in Appendix D.2 and Table 2. Although there is no significant difference in the results of Table 2, we believe that constructing more effective pseudo samples for training is an exciting and promising direction to further improve the learning ability and generalizability of language models, which is still under exploration.

---

### Author Response · Authors · 2022-08-02
**General Response to All Reviewers**

We thank the reviewers for their insightful and constructive reviews. We first provide responses to several points raised by multiple reviewers. Responses to individual reviewers are provided below. **The Appendix that contains additional experiments is attached in the Supplementary Materials as a separate pdf file.**

**Design of the loss function (Reviewers 95eQ and HDLP)**

Thanks for the insightful question. Indeed, we have tried other loss functions
such as the MSE loss and the margin loss, and the experiments of other loss functions
can be found in Appendix C.
In particular, we have explained why the margin loss achieves inferior results. When trained with margin loss, -log( 1 - max(cur - alpha * prev, 0 )), the model is encouraged to learn a smaller probability than that of the last repetition. As shown in Figure 8,  in order to minimize the margin loss, the model learns a ‘cheap’ solution, in which the probability of repeating the previous sentence is directly reduced to almost 0,
regardless of the probability of the previous repetition. This clearly causes over-penalization: the model quickly reduces TP to close to 0 even if there is only one sentence repetition. Hence, the margin loss leads to worse performance than our loss function.

The motivation of our loss function (Eqn.1) is two-fold: 1) cur should not be much larger than alpha * prev to avoid the self-reinforcement effect; 2) cur should not be much smaller than alpha * prev to avoid over-penalization that directly reduces cur to an extremely low value as shown in Figure 8. Another possible choice is to restrict the value of cur to be in [lower\_bound, pre*cur] to avoid over-penalization, it introduces an extra hyper-parameter
lower\_bound. Empirically, we find that the loss function in Eqn.1 works the best.

**Regarding our explanation for why the sentence repetition occurs (Reviewer HDLP and tE1F)**

In this manuscript, we view the language model $P_{\theta}(\cdot|\mathbf{x}_{<t})$ as a black box, investigate the relationship between the probability of the token and the number of repetitions in context, and propose a training-based method, DITTO, to avoid self-reinforcement effect and mitigate repetitions.
Our main observations, and our explanation for why the sentence repetition occurs
can be summarized as follows:

 * Through our quantitative experiments across various corpus, we have several interesting observations:
    - ob1: The model tends to raise the probability of repeating the previous sentence. Specifically, even if there is only one sentence-level context repetition, the probability of repetition at the current step increases in most cases.
    - ob2: Self-reinforcement effect: the probability of repetition increases almost monotonically with the number of historical repetitions.
    - ob3: Sentences with higher initial probabilities usually have a stronger self-reinforcement effect.
* Furthermore, combined with maximization-based decoding algorithms such as greedy decoding, we provide explanations about the reasons why sentence repetition occurs
    - Since previous sentences have a high likelihood (generated by greedy search), they have more potential to be repeated according to ob1 and ob3.
    - Once the model repeats the sentence several times, the model would get stuck in the sentence loop due to the self-reinforcement effect according to ob2.

We admit that there must be deeper reasons for why the model raises the probability of repeating tokens from the perspective of model embedding, neural network architecture or intrinsic characteristics of language. Our current analysis has not touched these aspects and hence we do not claim to have a deep and comprehensive explanation of this phenomenon. Several such directions are still under exploration and we leave them as important future work. To avoid over-claiming, we will discuss it in Sec.6.

---

### Meta-Review · Area_Chair_2NqR · 2022-08-21

**Recommendation:** Accept
**Confidence:** Certain

**Metareview:**

This paper investigates the source of repetition in text generation from a language model and presents a training method to mitigate this problem. Their experiments show the proposed method not only reduces repetition, but also improves generation quality. I think this is a good paper. All reviewers agreed with me so I recommend acceptance to NeurIPS.

**Award:**

No

---

### Decision · Program_Chairs · 2022-09-14

Accept